

# Are lower levels of physical activity and self-rated fitness associated with higher levels of psychological distress in Croatian young adults? A cross-sectional study

Lovro Štefan[1], Goran Sporiš[1,]* and Tomislav Krističević[2,]*

[1] Department of General and Applied Kinesiology, Faculty of Kinesiology, University of Zagreb, Zagreb, Croatia
[2] Department of Sports Kinesiology, Faculty of Kinesiology, University of Zagreb, Zagreb, Croatia
* These authors contributed equally to this work.

Corresponding author
Lovro Štefan,
lovro.stefan1510@gmail.com

## ABSTRACT

**Background:** Although previous evidence has shown that physical activity and physical fitness lower the level of psychological distress, little is known of simultaneous associations between of physical activity and physical fitness and with psychological distress, especially in young adults. Therefore, the main purpose of the present study was to explore both separate and simultaneous association between physical activity and physical fitness with psychological distress.

**Methods:** Participants in this cross-sectional study were 2,100 university students (1,041 men and 1,059 women) chosen from eight faculties in the city of Zagreb. Physical activity, physical fitness and psychological distress were assessed using structured questionnaires. The associations were examined using logistic regression analysis.

**Results:** After adjusting for gender, body-mass index, self-rated health, material status, binge drinking, chronic disease/s and sleep quality, "insufficient" physical activity (OR = 2.60; 95% CI [1.92–3.52]) and "lower" levels of physical fitness (tertile 2; OR = 1.94; 95% CI [1.25–3.01] and tertile 1; OR = 2.59; 95% CI [1.65–4.08]) remained associated with "high" psychological distress. When physical activity and physical fitness were entered simultaneously into the model, "insufficient" physical activity (OR = 2.35; 95% CI [1.72–3.21]) and "lower" levels of physical fitness (tertile 2; OR = 1.77; 95% CI [1.24–2.77] and tertile 1; OR = 2.00; 95% CI [1.26–3.20]) remained associated with "high" psychological distress.

**Conclusion:** Our study shows that both "insufficient" physical activity and "lower" levels of physical fitness are associated with "high" psychological distress, even after adjusting for numerous covariates. Therefore, special policies aiming to increase the levels of physical activity and fitness are warranted.

## INTRODUCTION

Mental health disorders have become one of the main public health problems worldwide, with special increasing prevalence among youth (*Costello et al., 2003*).

Approximately 30% of children and adolescents present mental disorders in the United States (*Costello et al., 2003*). In Croatia, the prevalence of mental disorders among youth is 15.7% (*Rudan et al., 2005*), girls experienced high psychological distress more frequently in the last 30 days, compared with boys (33% vs. 16%) (*Novak & Kawacki, 2015*). In general, psychological distress is a term frequently used to describe the experience of unpleasant emotions and feelings that influence on everyday functioning (*Perales, del Pozo-Cruz & del Pozo-Cruz, 2014*) and has been consistently associated with cardiovascular (*Mensah & Collins, 2015*), metabolic (*Newcomer, 2007*) and musculoskeletal (*Patten, Williams & Wang, 2006*) diseases, cancer (*Kisely, Crowe & Lawrence, 2013*) and overall mortality (*Walker, McGee & Druss, 2015*).

Treatment for depression is antidepressant medication (*Richey & Krystal, 2011*), which is expensive with potential side effects. Nevertheless, regular physical activity may serve as a protective factor against mental disorders. Physical activity is defined as "any bodily movement produced by skeletal muscles that results in energy expenditure and can be categorized into occupation, sports, conditioning, household, or other activities" (*Caspersen, Powell & Christenson, 1985*). The health benefits of physical activity on mental health have been well-documented (*World Health Organization, 2010*). Specifically, *Larun et al. (2006)* in their meta-analytical review showed, that vigorous physical activity had small effect in reducing anxiety or depression symptoms in youth. One meta-analysis showed that overall effects of physical activity on mental health were small, yet significant and indicated that physical activity led to improvements of mental health outcomes in children (*Ahn & Fedewa, 2011*). To determine causal direction, one longitudinal study showed that the number of hours spent in physical activity per week at age 15–16 was negatively associated with emotional symptoms in boys, yet no associations were found in girls (*Sagatun et al., 2007*). The same study also showed, that boys and girls who spent 5–7 h of physical activity per week at the same age had the least mental difficulties reported after a three-year of follow-up (*Sagatun et al., 2007*). While physical activity is related to the body movement, physical fitness represents "a set of attributes that people have or achieve" (*Caspersen, Powell & Christenson, 1985*). In general, physical fitness has two components: (1) health-related and (2) skill-related, where each component has a set of sub-components (*Caspersen, Powell & Christenson, 1985*). Similar studies aiming to explore the associations between physical fitness and psychological distress have also shown, that young adults with higher levels of cardio-respiratory, strength and flexibility fitness (*Pozuelo-Carrascosa et al., 2017*; *Jeoung, Hong & Lee, 2013*) have significantly lower levels of mental health disorders, compared to their peers with lower levels of fitness and higher levels of mental health disorders.

Previous studies have tried to explain the complexity and mutual processes of biological and psychological factors that physical activity has on mental health (*Faulkner & Taylor, 2009*). In terms of biological factors, physical activity has beneficial effects on neurotransmitters (monoamines, dopamine, endorphin), which play an important role in regulating stress and emotions and rewarding motivation (*Paluska & Schwenk, 2000*). Psychological factors include self-esteem, self-efficacy and distraction,

and physical activity/fitness play an important role in increasing such perceptions (*Ekeland et al., 2004*).

Thus, according to the literature, it is well-established, that both physical activity and physical fitness have beneficial role on mental health. However, little is known about the relationship these two potential factors with mental health in young adults. Young adulthood is characterized by the onset of which mental health problems start to occur (*Kessler et al., 2005*). On the other hand, it has been reported that 40% of young adults from United States do not meet the recommendations of National physical activity guidelines (150 min of moderate or 75 min of vigorous physical activity weekly) leading to excessive weight gain (*Malhotra et al., 2013*) and other diseases (*Warburton, Nicol & Bredin, 2006*).

Since young adults represent a risk group for higher levels of mental health disorders and lower levels of physical activity, it is necessary to explore these associations. Therefore, the main purpose of the present study was to explore both separate and simultaneous associations of physical activity and physical fitness with psychological distress in a large sample of young adults.

## MATERIALS AND METHODS

### Participants

This study was conducted among university students in Zagreb, the capital city of Croatia with approximately 1,000,000 citizens. The University of Zagreb is composed of 33 faculties (departments) and between 65,000 and 70,000 students attend the University every year. A random sampling approach was used to select faculties. At the first stage, we randomly selected eight out of 33 faculties. The randomization was done with replacement, where each faculty had unique number and was drawn from the box. At the second stage, we contacted teachers from each faculty to help us organize the sampling procedure. A recruitment announcement was sent via emails and e-newsletter to the teachers with a request to pass the study information to students. All eight faculties agreed to take part in the study, representing 2,320 students enrolled in the 2017 academic year. Of these, 2,100 students (1,041 men and 1,059 women, aged 18–24 years) provided full data (90.5%) and were enrolled in further analysis. Students came from a variety of social (psychology, political sciences, economy and business), technical (computing, information technologies, electrical engineering, civil engineering, mechanical engineering, graphics arts and naval architecture) and health-related (medical doctors, physiotherapists, nurses) sciences. Before the main analysis, we examined the differences between the participants and non-participants in terms of gender, age, body-mass index, self-rated health and psychological distress. No significant differences were observed and no potential bias was made ($p = 0.21$–$0.74$). All the analysis and procedures were anonymous and in accordance with the Declaration of Helsinki and approved by the Institutional Review Board of the Faculty of Kinesiology (Ethics code: 16/2017). Also, all participants gave their written informed consent for participation in the study.

## Outcome variable

Psychological distress was assessed by using Kessler's six-item questionnaire: (1) "How often during the past 30 days did you feel nervous?," (2) "How often during the past 30 days did you feel hopeless?," (3) "How often during the past 30 days did you feel restless or fidgety?," (4) "How often during the past 30 days did you feel so depressed that nothing could cheer you up?," (5) "How often during the past 30 days did you feel that everything was an effort?" and (6) "How often during the past 30 days did you feel worthless?" (*Kessler et al., 2003*). Each question is scored from zero (none of the time) to four (all of the time). Scores of each question are summed up ranging from zero to 24, with lower score indicating lower level of psychological distress. *Kessler et al. (2003)* showed that responses <13 points vs. ≥13 points discriminated participants without and with psychological distress.

## Physical activity

To assess physical activity in the last seven days, we used International Physical Activity questionnaire, a reliable and valid instrument designed to measure physical activity in respondents between ages 18 and 65 (*Craig et al., 2003*). Specifically, this measure assesses the types of intensity of physical activity during the day to estimate total physical activity measured in metabolic equivalent units-min/week. We created a dichotomized variable, where "sufficiently active" participants participated in at least (1) 150 min/week in moderate physical activity or (2) 75 min of vigorous physical activity or (3) an equivalent combination of both compared with "insufficiently active" participants (*World Health Organization, 2010*).

## Physical fitness

Self-rated physical fitness was assessed by using one-time question: "How would you rate your physical fitness?" ranging from one (very poor) to 10 (excellent) (*Plante, LeCaptain & McLain, 2000*). This measure has previously been correlated with measures of objective physical fitness and perceived well-being (*Plante, Lantis & Checa, 1998*) and used in similar studies (*Gerber et al., 2010*).

## Covariates

Previous studies have shown, that body-mass index, socioeconomic status, alcohol consumption, having a long-term health condition, self-rated health and sleep quality are known or suspected to influence psychological distress and we included them as covariates in the analysis (*Novak & Kawacki, 2015*; *Perales, del Pozo-Cruz & del Pozo-Cruz, 2014*; *Sagatun et al., 2007*). Participants self-reported their height in meters (m) and weight in kilograms (kg), from which body-mass index ($kg/m^2$) was calculated. Before the study began, we had chosen 35 men and 40 women to validate self-reported height and weight with the objective measure taken by trained survey staff. Pearson's coefficient of correlation showed excellent relationship between two measures in men ($r = 0.96$) and women ($r = 0.97$). For the purpose of this study, we divided body-mass index score into two categories: (1) normal (<25 $kg/m^2$) vs. (2) overweight/obesity (≥25 $kg/m^2$).

Although not appropriate as a clinical tool, self-reported BMI serves as a valid tool for epidemiological surveys, especially in young adults (*Meyer et al., 2012*). Self-rated health was assessed using one-item question: "How would you rate your health?" Answers were arranged along a Likert-type scale as follows: (1) very poor, (2) poor, (3) fair, (4) good and (5) excellent. For the purpose of this study, we dichotomized the outcome variable into "good" (fair, good and excellent) vs. "poor" (very poor and poor) self-rated health (*Štefan et al., 2017*). Material status was assessed by one question: "How would you perceive your material status, based on your parents' occupation?" Responses were: (1) low, (2) medium and (3) high. We created two categories as follows: (1) low and (2) medium/high. Binge alcohol consumption was assessed by one question: "How often do you have (for men) five or more and (for women) four or more drinks on one occasion?" (*Peltzer & Pengpid, 2016*). Those who had (for men) five or more and (for women) four or more drinks on one occasion were categorized as "Yes," compared to "No" group who had less drinks on one occasion. The presence or absence of a chronic disease was asked by one-item question: "Have you ever been told by a doctor, that you suffer from any kind of chronic disease?" with "Yes" and "No" answers. To assess sleep quality, we asked about current self-perceived state of sleep quality: "How would you perceive your sleep quality?" Answers were arranged across a four-item scale as follows: (1) very good, (2) good, (3) poor and (4) very poor. Very good and good collapsed into "good" and poor and very poor into "poor" sleep quality.

## Data analysis

Basic descriptive statistics of the study participants are presented as frequencies (*N*) and percentages (%). Differences between categorical variables between "low" and "high" psychological distress were analyzed using Chi-square test. To explore the associations between physical activity and physical fitness with psychological distress, we performed a set of logistic regression analyses. We calculated odd ratios (ORs) with 95% confidence intervals (95% CIs). "High" psychological distress was the main outcome of the present study. First, we explored the association between "insufficient" physical activity and "high" psychological distress in model 1. Second, we explored the association between "low" physical fitness and "high" psychological distress. Since physical fitness was assessed by a 10-item scale, we calculated median and interquartile range (25th and 75th percentile range), in order to categorize participants into three groups (tertiles): (1) <25th percentile as "low," (2) 25th–75th percentiles as "medium" and (3) >75th percentile as "high" physical fitness. Finally, we entered both physical activity and physical fitness simultaneously into the model (model 3). Significance was set up at $\alpha \leq 0.05$ and it was one-sided. All the analysis were performed in Statistical Package for Social Sciences Software, ver. 22 (IBM Corp., Armonk, NY, USA).

## RESULTS

Basic descriptive statistics of the study participants are presented in Table 1. The prevalence of "high" psychological distress was 10.6%, while 22.6% and 28.2% of the study participants were "insufficiently" active and were in the "lowest" physical
**Table 1 Basic descriptive statistics of the study participants, Croatia (2017).**

| Study variables | Total sample (N = 2,100) | Low psychological distress (N = 1,878) | High psychological distress (N = 222) | p-value* |
|---|---|---|---|---|
| | N (%) | N (%) | N (%) | |
| **Physical activity** | | | | |
| Sufficient | 1,626 (77.4) | 1,502 (92.4) | 124 (7.6) | |
| Insufficient | 474 (22.6) | 376 (20.0) | 98 (80.0) | <0.001 |
| **Physical fitness** | | | | |
| Tertile 3 (highest) | 601 (28.6) | 572 (95.2) | 29 (4.8) | |
| Tertile 2 | 907 (43.2) | 809 (90.2) | 98 (10.8) | |
| Tertile 1 (lowest) | 592 (28.2) | 497 (84.0) | 95 (16.0) | <0.001 |
| **Gender** | | | | |
| Men | 1,041 (49.6) | 975 (93.7) | 66 (6.3) | |
| Women | 1,059 (50.4) | 903 (85.3) | 156 (14.7) | <0.001 |
| **Body-mass index** | | | | |
| Normal | 1,765 (84.0) | 1,577 (89.3) | 188 (10.7) | |
| Overweight/obesity | 335 (16.0) | 301 (89.9) | 34 (10.1) | 0.847 |
| **Self-rated health** | | | | |
| Good | 1,935 (92.1) | 1,750 (90.4) | 185 (9.6) | |
| Poor | 165 (7.9) | 120 (77.6) | 37 (22.4) | <0.001 |
| **Material status** | | | | |
| Middle/high | 2,048 (97.5) | 1,839 (89.8) | 209 (10.2) | |
| Low | 52 (2.5) | 39 (75.0) | 13 (25.0) | 0.002 |
| **Binge drinking** | | | | |
| No | 1,530 (72.9) | 1,391 (90.9) | 139 (9.1) | |
| Yes | 570 (27.1) | 487 (85.4) | 83 (14.6) | <0.001 |
| **Chronic disease/s** | | | | |
| No | 1,905 (90.7) | 1,721 (90.3) | 184 (9.7) | |
| Yes | 195 (9.3) | 157 (80.5) | 38 (19.5) | <0.001 |
| **Sleep quality** | | | | |
| Very good/good | 687 (32.7) | 658 (95.8) | 29 (4.2) | |
| Poor/very poor | 1,413 (67.3) | 1,220 (86.3) | 192 (13.7) | <0.001 |

**Note:**
* Chi-square test.

fitness group. We found that higher percentage of "insufficiently" active participants and those with "lower" levels of physical fitness reported having "high" psychological distress in the last 30 days. Also, higher percentage of women, those participants who reported having poor self-rated health, low material status, binge drinking, having a chronic disease and poor/very poor sleep had "high" psychological distress in the last 30 days.

The associations between physical activity, physical fitness and psychological distress are presented in Table 2. In model 1, "insufficient" physical activity (OR = 2.60; 95% CI [1.92–3.52]) was associated with "high" psychological distress. In model 2, those participants in lower tertiles (tertile 2; OR = 1.94; 95% CI [1.25–3.01] and tertile 1; OR = 2.59; 95% CI [1.65–4.08]) were more likely to experience "high" psychological distress.

**Table 2 Odd ratios for high psychological distress of the study participants, Croatia (2017).**

| Study variables | Physical activity and psychological distress | Physical fitness and psychological distress | Physical activity and physical fitness with psychological distress |
|---|---|---|---|
| | OR (95% CI) | OR (95% CI) | OR (95% CI) |
| **Physical activity** | | | |
| Sufficient | 1.00 | | 1.00 |
| Insufficient | 2.60 (1.92–3.52)*** | | 2.35 (1.72–3.21)*** |
| **Physical fitness** | | | |
| Tertile 3 (highest) | | 1.00 | 1.00 |
| Tertile 2 | | 1.94 (1.25–3.01)** | 1.77 (1.14–2.77)* |
| Tertile 1 (lowest) | | 2.59 (1.65–4.08)*** | 2.00 (1.26–3.20)** |
| **Gender** | | | |
| Men | 1.00 | 1.00 | 1.00 |
| Women | 2.18 (1.57–3.02)*** | 2.17 (1.57–3.01)*** | 2.00 (1.44–2.78)*** |
| **Body-mass index** | | | |
| Normal | 1.00 | 1.00 | 1.00 |
| Overweight/obesity | 1.09 (0.71–1.67) | 1.01 (0.66–1.55) | 1.02 (0.66–1.57) |
| **Self-rated health** | | | |
| Good | 1.00 | 1.00 | 1.00 |
| Poor | 2.13 (1.38–3.30)*** | 2.03 (1.31–3.12)*** | 2.04 (1.31–3.16)*** |
| **Material status** | | | |
| Middle/high | 1.00 | 1.00 | 1.00 |
| Low | 2.32 (1.14–4.74)* | 2.12 (1.04–4.33)* | 2.22 (1.08–4.53)* |
| **Binge drinking** | | | |
| No | 1.00 | 1.00 | 1.00 |
| Yes | 1.71 (1.26–2.33)*** | 1.76 (1.30–2.38)*** | 1.75 (1.28–2.93)** |
| **Chronic disease/s** | | | |
| No | 1.00 | 1.00 | 1.00 |
| Yes | 1.91 (1.26–2.89)*** | 1.96 (1.30–2.95)*** | 1.94 (1.28–2.93)** |
| **Sleep quality** | | | |
| Very good/good | 1.00 | 1.00 | 1.00 |
| Poor/very poor | 3.44 (2.28–5.19)*** | 3.34 (2.22–5.04)*** | 3.29 (2.18–4.98)*** |

Notes:
Model 1: Examines the associations of physical activity with psychological distress adjusted for gender, body-mass index, self-rated health, material status, binge drinking, chronic disease/s and sleep quality.
Model 2: Examines the associations of physical fitness with psychological distress adjusted for gender, body-mass index, self-rated health, material status, binge drinking, chronic disease/s and sleep quality.
Model 3: Examines the associations of physical activity and physical fitness entered simultaneously into the model with psychological distress adjusted for gender, body-mass index, self-rated health, material status, binge drinking, chronic disease/s and sleep quality.
* $p < 0.05$.
** $p < 0.01$.
*** $p < 0.01$.

Finally, when both physical activity and physical fitness were entered simultaneously into the model (model 3), OR for "insufficient" physical activity (OR = 2.35; 95% CI [1.72–3.21]) and for "lower" levels of physical fitness (tertile 2; OR = 1.77; 95% CI [1.14–2.77] and tertile 1; OR = 2.00; 95% CI [1.26–3.20]) decreased, but remained associated with "high" psychological distress. All three models were adjusted for gender, body-mass index, self-rated

health, material status, binge drinking, chronic disease/s and sleep quality. The association between physical fitness and physical activity was moderate ($r = 0.33$, $p < 0.001$) and variance inflation factors test showed no multicollinearity (1.00–1.10).

## DISCUSSION

The main purpose of the present study was to explore both separate and simultaneous associations between physical activity and physical fitness and psychological distress in a large sample of young adults. Both "insufficient" physical activity and "lower" levels of physical fitness were associated with "high" psychological distress after adjusting for gender, body-mass index, self-rated health, material status, binge drinking, chronic disease/s and sleep quality.

Our results are in line with previous studies aiming to explore the associations between physical activity/fitness and mental health (*Sagatun et al., 2007*; *World Health Organization, 2010*; *Larun et al., 2006*; *Pozuelo-Carrascosa et al., 2017*; *Jeoung, Hong & Lee, 2013*). Specifically, *Sagatun et al. (2007)*, in their three-year longitudinal study, showed that weekly hours of physical activity were negatively associated with emotional symptoms or peer problems only in boys, but not in girls. As mentioned before, mental health is a complex state comprised of behavioral, psychological and social components (*Paluska & Schwenk, 2000*; *Ekeland et al., 2004*; *Bandura, 1977*). A few previous studies have shown, that participants, who are engaged in regular physical activity displays much less inhibition in social behavior (*Kirkcaldy, Shephard & Siefen, 2002*) and scored lower on psychological discomfort captured by loneliness, hopelessness and shyness (*Page & Tucker, 1994*) compared to their physically inactive peers. In addition to physical activity, studies aiming to explore the associations between physical fitness and mental health disorders have shown that higher levels of cardio-respiratory, strength and flexibility fitness are significantly associated with decreased mental health disorders, compared to participants with lower levels of physical fitness (*Pozuelo-Carrascosa et al., 2017*; *Jeoung, Hong & Lee, 2013*). Our results confirmed strong negative association between self-perceived physical fitness and psychological distress, that is, "lower" levels of physical fitness were associated with "high" psychological distress.

The mechanism underlying the association between physical activity/fitness and mental health is not clearly understood (*Blake, 2012*). Previous meta-analytical reviews have shown small clinical effect that physical activity has on mental health (*Larun et al., 2006*; *Ahn & Fedewa, 2011*; *Rimer et al., 2012*). Although small, positive benefits of physical activity on health outcomes have been well-documented (*Warburton, Nicol & Bredin, 2006*). In terms of mental health, physical activity serves as beneficial factor for neurotransmitters in the brain, leading to increased levels of motivation and positive emotions and reducing stress and pain (*Paluska & Schwenk, 2000*; *Ekeland et al., 2004*; *Bandura, 1977*). Our results also showed that among numerous factors we adjusted for, the strongest association was between "very poor/poor" sleep quality and "high" psychological distress, which is similar to other studies (*Feng et al., 2014*). Such associations between sleep quality and psychological distress could be mediated by physical activity which regulates temperature following exercise and the onset of sleep

declines through vasodilatation of peripheral heat dissipation (*Driver & Taylor, 2000*). In that way, by affecting on sleep quality, physical activity decreases psychological distress and improves behavioral and emotional regulations.

Our study has some limitations. First, we used a cross-sectional design, in order to determine the associations between physical activity and physical fitness with psychological distress. To determine the causality, *Reichenheim & Coutinho (2010)* reported that the main outcome of the study should be frequent and might be different among subjects, due to a dynamic population. Second, we used subjective measures to assess psychological distress, physical activity, physical fitness and other covariates. However, self-reported measures are largely used in epidemiological studies. But, for better precision, future studies should use direct measurement method (motor and functional fitness tests) over a longer period of time, in order to track and establish causal direction of the association between physical activity/fitness and psychological distress.

## CONCLUSION

Our results show strong associations between "insufficient" physical activity and "lower" levels of physical fitness with "high" psychological distress in a large sample of young adults. Findings of this study should be taken into account when establishing and implementing special strategies and policies that leverage higher participation in physical activity in order to decrease "high" psychological distress.

## ACKNOWLEDGEMENTS

We would like to thank teachers and students for their enthusiastic participation in the study.

### Funding
The authors received no funding for this work.

### Competing Interests
Goran Sporiš is an Academic Editor for PeerJ.

### Author Contributions
- Lovro Štefan conceived and designed the experiments, performed the experiments, analyzed the data, contributed reagents/materials/analysis tools, prepared figures and/or tables, authored or reviewed drafts of the paper, approved the final draft.
- Goran Sporiš conceived and designed the experiments, analyzed the data, prepared figures and/or tables, authored or reviewed drafts of the paper, approved the final draft.
- Tomislav Krističević conceived and designed the experiments, analyzed the data, prepared figures and/or tables, authored or reviewed drafts of the paper, approved the final draft.

## Ethics

The following information was supplied relating to ethical approvals (i.e., approving body and any reference numbers):

All the analysis and procedures were anonymous and in accordance with the Declaration of Helsinki and approved by the Institutional Review Board of the Faculty of Kinesiology (Ethics code: 16/2017).

## Data Availability

The raw data are provided as a Supplemental File.

## Supplemental Information

Supplemental information for this article can be found online at http://dx.doi.org/ 10.7717/peerj.4700#supplemental-information.

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
