# Peer review of "Are lower levels of physical activity and self-rated fitness associated with higher levels of psychological distress in Croatian young adults? A cross-sectional study"

_PeerJ, doi:10.7717/peerj.4700_

## Round 0.1 · original submission · Major Revisions

The manuscript provides important information but requires changes including revising the English writing. In addition, please note one reviewer has attached an annotated manuscript to their review that has the detailed comments. Please Provide a point by point response with the revised manuscript denoting how each issue was addressed.

Reviewer 1 ·

Basic reporting

The manuscript is generally well written and good to read. Some observations were made directly into the file.

There are some citations lacking and some parts of the Methods were not presented, which was pointed out in the text presenting Tables.

Other problems, information and questions were indicated in the pdf.

Experimental design

Observational study.

Validity of the findings

See the pdf.

Additional comments

Review needed.

Annotated reviews are not available for download in order to protect the identity of reviewers who chose to remain anonymous.

·

Basic reporting

The text should be revised throughout as there are some English Language issues.

The text is sufficiently well referenced but there the authors should consider elaborating more on physical fitness and physical activity

The raw data is clear and well described

The objective and hypothesis is clear

Experimental design

This is a good research involving a population-based study on a very specific topic.

The investigation was performed to a highly professional and ethical standard

There is room for improving the description of the methods used (for example the questionnaires...)

Validity of the findings

The data is statistically sound and robust

The conclusion section answers the hypothesis question well.

Additional comments

First of all, it is a pleasure to read this article built on a population-based study investigating the relationship between physical activity and fitness and psychological distress. The following points should be considered to make this research article more robust.

1. The English language used in this manuscript should be improved so as the readers can understand clearly your text. Some examples where language could be improved are lines 83 to 86; line 96, line 119 (65000 to 70000 students...), line 145 (...ranging from 0 to 24). The text should be thoroughly revised to check for grammatical errors such as line 260(possibility..)

2. There is a big chunk of repetition of the last sentence in discussion (lines 264 to 266) and conclusions (272 to 274)..Consider rather to elaborate this sentence further in discussion and explain more the ALPHA battery fitness objective method, how it is better than the previously used method(s).

3. Authors posited that there was no significant difference in line 132 but does not explain clearly differences among which factor(s)...same need to explain the statistics and which variable when they mentioned the two measures in line 164 to 165.

4. The authors should consider elaborating what is physical activity and physical fitness as well as psychological stress further in the text (introduction) and distinguish between physical activity and physical fitness.

5. Overall the text seem fine with clear objectives and clear conclusions as well as good presentation of statistical result in tabular forms.

6. I thank the authors for the providing the raw data with clear variable names for each column of data.

---

## Round 0.2 · accepted · Accept

The manuscript has been sufficiently revised and reviewers are satisfied with the corrections that have been made.

# ·

Basic reporting

The text is clear and the English issues are addressed.

The text is sufficiently well referenced.

The raw data is clear and well described.

The objective and hypothesis is clear.

Experimental design

This is a good research involving a population-based study on a very specific topic.

The investigation was performed to a highly professional and ethical standard.

Validity of the findings

The data is statistically sound and robust as well as the conclusion section answers the hypothesis question well.

Additional comments

First of all, it is a pleasure to read this article built on a population-based study investigating the relationship between physical activity and fitness and psychological distress. Overall the text seem fine with clear objectives and clear conclusions as well as good presentation of statistical result in tabular forms. I thank the authors for the providing the raw data with clear variable names for each column of data.